# Nanoparticle Growth by Particle Phase Chemistry

Michael J. Apsokardu and Murray V. Johnston

Department of Chemistry and Biochemistry, University of Delaware, Newark, DE, 19716, USA

*Correspondence to*: Murray V. Johnston (mvj@udel.edu)

**Abstract**

The ability of particle phase chemistry to alter the molecular composition and enhance the growth rate of nanoparticles in the 2-100 nm diameter range is investigated through the use of a growth model. The molecular components included are sulfuric acid, ammonia, water, a non-volatile organic compound, and a semi-volatile organic compound. Molecular composition and growth rate are compared for particles that grow by partitioning alone vs. those that grow by a combination of partitioning and an accretion reaction in the particle phase between two organic molecules. Particle phase chemistry causes a change in molecular composition that is particle diameter dependent, and when the reaction involves semi-volatile molecules, the particles grow faster than by partitioning alone. These effects are most pronounced for particles larger than about 20 nm in diameter. The modeling results provide a fundamental basis for understanding recent experimental measurements of the molecular composition of secondary organic aerosol showing that accretion reaction product formation increases linearly with increasing aerosol volume-to-surface area. They also allow initial estimates of the reaction rate constants for these systems. For secondary aerosol produced by either OH oxidation of the cyclic dimethylsiloxane ($D_5$) or ozonolysis of β-pinene, oligomerization rate constants on the order of $10^{-3}$ to $10^{-1}$ $M^{-1}s^{-1}$ are needed to explain the experimental results. These values are consistent with previously measured rate constants for reactions of hydroperoxides and/or peroxyacids in the condensed phase.

## 1 Introduction

Atmospheric aerosols influence Earth's energy balance either directly by scattering incoming solar radiation or indirectly through cloud formation (Charlson et al., 1992; Kerminen et al., 2012; Lohmann and Feichter, 2004). A significant fraction of airborne particles originate from gas-to-particle conversion during new particle formation (NPF) (Bzdek and Johnston, 2010). Volatile compounds emitted into the atmosphere from biogenic and anthropogenic sources are oxidized to produce semi-volatile and non-volatile products. Particle formation begins when these products, in combination with other gas phase species, come together to form clusters on the order of 1-2 nm in diameter that are able to spontaneously grow to much larger sizes (Bzdek and Johnston, 2010; Dusek et al., 2006; Kulmala et al., 2013; Zhang et al., 2012). Depending on chemical composition, once the particles have grown to a size of 50-100 nm in diameter, they are able to serve as cloud condensation nuclei (CCN). The probability that a newly formed nanoparticle will grow to a CCN relevant size depends on its growth rate relative to the loss rate from e.g. coagulation or scavenging by pre-existing aerosols (Kuang et al., 2010). Uncertainties in predicting the

conditions that favor CCN formation make it challenging to accurately predict future impacts of radiative forcing (Carslaw et al., 2013).

The three main chemical species that contribute to ambient nanoparticle growth are sulfuric acid, a neutralizing base typically ammonia, and organic matter.  The growth rate due to sulfuric acid along with neutralizing base is accurately predicted by experimental measurements of gas phase mixing ratio and particle phase composition using a condensational growth model (Bzdek et al., 2013; Pennington et al., 2013; Smith et al., 2008; Stolzenburg et al., 2005), though sulfuric acid represents only a minor fraction of the total growth rate of ambient particles (Kuang et al., 2010, 2012; Weber et al., 1996; Wehner et al., 2005). Nanoparticle composition and growth rate are dominated by organic matter (Bzdek et al., 2011, 2012, 2013, 2014a, 2014b; Pennington et al., 2013; Riipinen et al., 2012; Smith et al., 2008), and though significant molecular insight has been gained (Bianchi et al., 2016; Ehn et al., 2014; Kulmala et al., 2013; Riccobono et al., 2014), current growth models for organic matter appear to be incomplete (Hallquist et al., 2009; Tröstl et al., 2016).

Gas phase oxidation of volatile organic compounds occurs through a complex set of reaction pathways in both the gas and particle phases to yield particle phase products that often number in the hundreds or thousands based on accurate mass measurements (Bateman et al., 2008; Mentel et al., 2015; Reinhardt et al., 2007; Tu et al., 2016).  Absorptive partitioning (Barsanti et al., 2017; Pankow, 1994) of gas phase products to the particle phase forms secondary organic aerosol (SOA), which includes both non-volatile (NVOC) and semi-volatile (SVOC) organic compounds (Kroll and Seinfeld, 2008; Riipinen et al., 2012).  NVOC have a negligible evaporation rate after partitioning to the particle phase, and therefore cause particles to grow at a rate given by the condensation rate. SVOC have a significant evaporation rate from the particle phase, and therefore grow particles at a slower rate than the condensation rate. Recent advances in measurement technology have greatly improved our molecular understanding of NVOC (Ehn et al., 2012, 2014; Jokinen et al., 2015).  More broadly, the complex distribution of molecular products and their associated wide range of volatilities can be represented by a volatility basis set (VBS) distribution where products are binned according to their gas phase saturation concentrations (Donahue et al., 2006, 2011, 2012).

Molecular analysis of SOA has shown the presence of compounds that were produced by a reaction within the particle phase.  Some examples include oligomers in biogenic SOA formed by accretion reactions (Barsanti and Pankow, 2004; Kalberer et al., 2004; Tolocka et al., 2004a), imine related species formed by the reaction of dicarbonyls with ammonia or amines (Galloway et al., 2014; De Haan et al., 2011; Lee et al., 2013; Stangl and Johnston, 2017), and organosulfates (Riva et al., 2016; Surratt et al., 2007; Wong et al., 2015; Xu et al., 2015). Reactions such as these increase the aerosol yield by forming additional SOA beyond what would be expected from partitioning alone, if they form non-volatile products from semi-volatile reactants in the particle phase (Lopez-Hilfiker et al., 2016; Shiraiwa et al., 2014).  Experimental measurements have shown that oligomers can constitute up to about 50% of the mass of SOA produced from biogenic precursors in laboratory reactors, though it is not clear how much of the oligomeric matter is produced from semi-volatile vs. non-volatile precursors (Hall IV and Johnston, 2011).

70   The role of particle phase oligomerization in SOA formation has been the focus of several modeling studies. Owing to the high molecular weight and corresponding low volatility of oligomer products (Shiraiwa et al., 2014), early work assumed an irreversible process (Vesterinen et al., 2007), which proved effective for predicting the yields of freshly formed aerosol in chamber experiments and estimating the magnitude of the oligomerization rate constant needed for the process to impact yields. More recent models have included reversibility (Roldin et al.,

75 2014; Trump and Donahue, 2014), which is needed to reproduce perturbations of freshly formed SOA such as changes induced by isothermal dilution, thermal degradation and/or aging. Regional air quality models show that oligomerization has the potential to significantly increase the SOA mass concentration (Aksoyoglu et al., 2011; Lemaire et al., 2016), and accurately representing this chemistry in these models is perhaps the greatest uncertainty for predicting SOA formation (Shrivastava et al., 2016).

80   The influence of particle size, or more precisely the relative roles of particle volume and surface area, on oligomer formation in SOA has received relatively little attention though these effects are implicit in all of the models. Particle size has been discussed primarily in the context of accumulation mode particles greater than 100 nm in diameter (Roldin et al., 2014; Shiraiwa et al., 2013; Vesterinen et al., 2007). The Vesterinen study did report aerosol yield functions from simulations starting with 20 nm diameter seed particles, and the results suggested that

85 particle phase chemistry could enhance growth rates under atmospherically relevant conditions.

   In the work described here, we explore computationally the impact of particle phase chemistry on both the growth rate and composition of nanoparticles (defined here as particles smaller than 100 nm in diameter). This work was inspired by recent composition studies of laboratory-generated secondary aerosol spanning several tens of nanometers in diameter, which showed systematic changes in composition with increasing particle size (Tu and

90 Johnston, 2017; Wu and Johnston, 2017). In the first set of calculations, chemical composition and growth rate are determined for particles between 2 and 100 nm assuming that growth occurs by partitioning alone. These results are compared to a second set of calculations where semi-volatile molecules in the particle phase are allowed to undergo an accretion reaction to produce a non-volatile product. Together, the two sets of calculations give insight into the reaction conditions and particle size range where particle phase chemistry is most likely to alter nanoparticle

95 composition and accelerate the growth rate. The modeling approach is similar to that used in previous studies, though the reaction conditions studied here more closely resemble those of an "open" laboratory reactor where aerosol flows into and out of the reactor (i.e. flow reactor), as opposed to a "closed" batch reactor which was the main focus of previous modeling work. The results are discussed in the context of recent size-resolved molecular composition measurements (performed with flow reactors) and the potential atmospheric impact of particle phase

100 chemistry.

**2 Model Description**

   The growth model used in this work includes sulfuric acid, ammonia, and SOA since these are the major components found in ambient nanoparticles during NPF (Bzdek et al., 2011, 2014a; Pennington et al., 2013; Smith

et al., 2008; Stolzenburg et al., 2005). Water is also included as predicted by the Extended Aerosol Inorganics Model (E-AIM) (Wexler and Clegg, 2002), and then corrected to account for the effects of particle surface curvature (Kreidenweis et al., 2005; Yli-Juuti et al., 2013). For simplicity, the gas phase precursors to SOA formation are represented by two specific compounds, one non-volatile (NVOC) and the other semi-volatile (SVOC). Particle phase chemistry involves a generic accretion reaction between two SVOC (or NVOC) molecules to give a dimer.

Table 1 gives relevant molecular parameters and gas phase mixing ratios. The gas phase mixing ratios are assumed to be constant over time (steady state) as might be achieved in a flow reactor, and the values chosen are typical of what might be observed during NPF in a boreal forest (Vestenius et al., 2014). Molecular volatility is expressed in terms of the saturation concentration, C*, which is the equilibrium vapor pressure in units of μg/m³ (Donahue et al., 2006).

Calculations begin with a 2 nm diameter particle consisting of sulfuric acid and ammonia in a 2:1 ratio of base to acid. From there, gas phase species partition to the particle phase based on their volatilities and gas phase mixing ratios, causing the particle to grow. The growth calculation is iterative. Starting from an initial particle of a given composition and volume ($V_{p,n}$), the increase in particle volume (and corresponding change in composition) is determined over a short time period for each molecular species separately, and the increases for the individual

components are summed to give the total volume of the particle at the end of the time period ($V_{p,n+1}$) along with its composition. The end point for the first time period is used as the start point for the second time period, and the process repeats. A schematic of calculation of work flow is given in Supporting Information Figure S1 along with additional aspects of the approach.

### 2.1 Partitioning

The extent to which a compound formed in the gas phase partitions to the particle phase is determined by its saturation ratio ($S_D$), which is the ratio of the gas phase mixing ratio to the saturation mixing ratio. The subscript "$D$" acknowledges that the saturation ratio depends in part on the radius of curvature of the particle surface. Accordingly, the first step in the calculation is to determine the Kelvin effect modified vapor pressure ($KEMP_D$) for all species based on the initial particle diameter. For the conditions used in this study, sulfuric acid and NVOC have

$S_D \gg 1$ for all particle diameters investigated, and as discussed previously, grow particles at their condensation rates. Ammonia, water, and SVOC have $S_D \ll 1$ and grow particles at rates slower than their condensation rates.

The collision rate for species $i$ determines the number of condensing molecules that can be taken into the particle during time period d$t$, thereby incrementing the particle volume by d$V_i$:

(1) $$\frac{dV_i}{dt} = \frac{c_i}{4} \gamma \pi D^2 C_{i,g} \beta_D V_{M,i}$$

where $c_i$ is the mean thermal velocity, $\gamma$ is the uptake coefficient, $D$ is the particle diameter, $C_{i,g}$ is the gas phase mixing ratio of species $i$, $\beta_D$ is the Fuchs-Sutugin correction factor for mass transport to a spherical particle with diameter $D$, and $V_{M,i}$ is the molar volume of species $i$. The equation for $\beta_D$ is given in Supporting Information. Values of d$V_i$ for condensational growth by sulfuric acid and NVOC are calculated from Eq. 1, with the inherent

assumptions that the surface accommodation coefficient is 1, and the particle diameter is small enough that gas
phase diffusion does not limit the condensation rate.

Equation 1 can also be used to calculate the uptake of semi-volatile species provided that the total amount
taken into the particle does not exceed the equilibrium end point. Ammonia uptake is determined from the number
of sulfuric acid molecules that have condensed. For a given time period d$t$, once the number of ammonia molecules
that have been taken into the particle equals twice the number of sulfuric acid molecules that were taken during the
same time period, no further ammonia uptake occurs. While dissolution of ammonia into water is possible, the
amount is negligible in comparison to ammonia uptake associated with condensation of sulfuric acid (Clegg et al.,
1998; Wexler and Clegg, 2002). Under the conditions used in this study, the condensation rate of ammonia is about
25 times greater than that of sulfuric acid, so the model assumes that stoichiometric uptake of ammonia is
instantaneous with sulfuric acid. Water uptake is determined from E-AIM based on the combined amounts of
sulfuric acid, ammonia, and NVOC that have been added during time period d$t$. Since the gas phase mixing ratio of
water is very high, equilibrium is assumed to be achieved instantaneously during the time period.

Individual volume increments d$V_i$ for sulfuric acid, NVOC, ammonia, and water are summed to give the total
volume increment. Because the volume of particle phase has increased, SVOC is no longer in equilibrium between
the gas and particle phases, and a net migration of SVOC from the gas phase to the particle phase must occur. SVOC
molecules are taken into the particle at a rate described by Eq. 1 to re-establish equilibrium. The equilibrium point
(Pankow, 1994) expressed as the volume ratio of SVOC in the particle phase ($V_i/V_{p,n}$) is:

$$(2) \qquad \frac{V_{i,}}{V_{p,n}} = \left(\frac{C_{i,g}}{KEMP_D}\right)\left(\frac{V_{M,i}}{V_{M,p}}\right)\frac{1}{\zeta_i}$$

where $i$ in this case refers to SVOC, $\zeta_i$ is the activity coefficient for SVOC in the particle phase (assumed to be 1 in
this study), and $V_{M,i}$ and $V_{M,p}$ are the respective the molar volumes of SVOC and the particle phase. The incremental
increase of SVOC over time period d$t$ is determined by evaluating Eq. 2 before and after the increases due to
sulfuric acid, NVOC, ammonia, and water. For the conditions used in this study, the condensation rate of gas phase
SVOC is generally greater than the uptake rate needed to maintain equilibrium when particle growth is restricted to
partitioning. The ratio of the two rates is dependent on the volume to surface area ratio of the particle, ranging from
several orders of magnitude for a 2 nm diameter particle to a factor of 200 for a 100 nm diameter particle. Because
the condensation rate is generally much greater than the uptake rate needed for partitioning, no evaporation rate of
particle phase SVOC was included in the current study. The one situation where condensation is not sufficient for
partitioning occurs when particle phase chemistry is considered and will be discussed later.

**2.2 Particle Phase Chemistry**

Particle phase chemistry is modeled as an irreversible accretion reaction where two monomers come
together to produce a non-volatile DIMER. While all combinations of SVOC and NVOC molecules were
considered, most calculations involved the reaction between two SVOC molecules:

(3)    $\frac{\mathrm{d}[DIMER]}{\mathrm{d}t} = -\frac{\mathrm{d}[SVOC]}{\mathrm{d}t} = k_{DIMER}[SVOC]^2$

where $k_{DIMER}$ is the second order rate constant, and *[SVOC]* is the particle phase concentration established by partitioning between the gas and particle phases. Since the reaction consumes particle phase SVOC molecules, additional SVOC must be taken from the gas phase into the particle phase to re-establish equilibrium. The rate at which additional SVOC molecules are taken into the particle is determined by the rate of particle phase reaction. When particle phase reaction is included in the growth calculation, the change in particle volume from $V_{p,n}$ to $V_{p,n+1}$ with respect to SVOC is the sum of the volume increase from unreacted SVOC molecules needed to re-establish partitioning equilibrium, and the volume increase from additional SVOC uptake to form dimers. SVOC continues to partition to the particle phase at each iteration, resulting in a continuous supply of reactant molecules for dimerization.

Dimerization rate constants in the range of $10^{-3}$ to $10^{-1}$ M$^{-1}$ s$^{-1}$ were studied, with most calculations at $10^{-2}$ M$^{-1}$s$^{-1}$, which is the rate constant reported for dimerization of glyoxal in a bulk aqueous solution (Ervens and Volkamer, 2010). Ziemann and Atkinson (Ziemann and Atkinson, 2012) have reviewed thermodynamic and kinetic data for several types of reactions relevant to biogenic SOA. The reaction of a hydroperoxide with a carbonyl to give a peroxyhemiacetal, and the reaction of a peroxyacid with a carbonyl to form an acyl peroxyhemiacetal, both have reported rate constants in the $10^{-4}$ to $10^{-2}$ M$^{-1}$s$^{-1}$ range depending on reaction conditions, and are relevant to the modeling results presented here. Reactions such as aldol condensation of carbonyls and ester formation from an acid and alcohol are much slower and unlikely to be atmospherically relevant based on both kinetics (Casale et al., 2007) and thermodynamics (DePalma et al., 2013). The effects of dimer decomposition (reaction reversibility), particle phase diffusion coefficient, and phase separation are not considered in this work, though we note that all would have the effect of reducing the impact of particle phase chemistry on composition and growth rate.

**3 Results and Discussion**

**3.1 Particle Growth by Partitioning**

The first set of calculations includes particle growth by partitioning, but no particle phase reaction. Figure 1a shows the diameter growth rate and Figure 1b the dry mass fraction for each chemical species as a function of particle diameter. Initially, the diameter growth rates for all species increase quickly with increasing particle size. For sulfuric acid and NVOC, the increase is most pronounced between 2 and 10 nm, and this size dependence arises directly from the Fuchs-Sutugin term that limits mass transport to small diameter particles. Above about 10 nm, the growth rates for these species become independent of particle diameter. A constant growth rate for non-volatile species has been noted previously (Weber et al., 1996) and can be understood by rewriting Eq. 1 in terms of the diameter growth rate:

(4)    $\frac{\mathrm{d}D}{\mathrm{d}t} = \frac{c_i}{2}\gamma C_{i,g}\beta_D V_{M,i}$

Above about 10 nm, $\beta_D$ becomes independent of $D$.  Since $\gamma$ is independent of $D$ for a surface-limited process such as condensation (Smith et al., 2003; Tolocka et al., 2004a), d$D$/d$t$ also becomes independent of $D$ provided that the precursor gas phase mixing ratio ($C_{i,g}$) does not change.  Ammonia uptake is driven by sulfuric acid uptake, and therefore follows the same particle diameter dependence.

Water and SVOC show greater increases in their growth rates with increasing particle diameter than
sulfuric acid and NVOC because they have $S_D < 1$.  In principle, semi-volatile species are subject to two particle size-dependent effects: $\beta_D$ in Eq. 1 and $KEMP_D$ in Eq. 2.  For the conditions studied here, the $\beta_D$ term has a negligible effect on the uptake rates of water and SVOC, since uptake is determined by the equilibrium endpoint rather than the condensation rate.   Instead, the difference between the semi-volatile species and non-volatile species in Figure 1a are driven by the particle size dependence of $KEMP_D$, which alters the equilibrium point for absorptive
partitioning.

In Figure 1b, the dry particle mass fractions of sulfuric acid, NVOC, and ammonia show very little change with increasing $D$ since their relative growth rates are independent of $D$.  Small changes just above 2 nm are due to the choice of starting composition of the 2 nm diameter particle.  In contrast, the SVOC mass fraction increases quickly with increasing particle size owing to the dependence of $KEMP_D$ on $D$.  Figure 2 shows the mass fraction
ratio of SVOC to NVOC as a function of particle size.  The ratio increases quickly at the smallest particle sizes and then more slowly thereafter.  This plot is consistent with experimental measurements of molecular composition across a similar range of particle diameters, which show that lower volatility species are preferentially detected in smaller particles (Winkler et al., 2012; Zhao et al., 2013).

### 3.2 Particle Growth by a Combination of Partitioning and Particle Phase Chemistry

When particle phase reaction is included in the growth calculation, both the growth rate and chemical composition change. Figures 3a and b show diameter growth rates and dry mass fractions, respectively, when particle phase reaction is included. Comparing Figure 3a to Figure 1a shows that particle phase chemistry causes the diameter growth rate to continue increasing with increasing particle size above 10 nm. This diameter dependence is different from the growth rate due to partitioning alone where d$D$/d$t$ becomes independent of particle size above 10
230     nm. The size dependence of particle growth at larger particle sizes can be understood by expressing the uptake coefficient in terms of particle diameter:

(5)     $\gamma = k'D$

In Eq. 5, the uptake coefficient is proportional to $D$, which is characteristic of a volume-limited process (Saul et al., 2006; Tolocka et al., 2004b).  Inserting Eq. 5 into Eq. 4, shows that d$D$/d$t$ also increases linearly with $D$.  The
number of SVOC molecules that react to form DIMER is proportional to the total volume of the particle, and therefore the SVOC uptake rate needed to maintain partitioning equilibrium is also volume (and hence diameter) dependent. Note that the SVOC and DIMER diameter growth rates are both proportional to $D$ in Figure 3a. Growth

by condensation of NVOC and sulfuric acid remain surface-limited, and therefore, these diameter growth rates are unaffected by dimer formation.

240   Figure 3b shows the dry mass fractions as a function of particle diameter when particle phase chemistry is included. Below about 10 nm, the plots in Figure 3b are identical to those in Figure 1b where partitioning alone is considered. Particle phase chemistry has minimal impact on the growth and composition of small particles for two reasons. First, the particle volume to surface area ratio is very small, which favors surface-limited processes (condensational growth by NVOC and sulfuric acid) over volume-limited processes (accretion reaction). Second,

245 the dependence of $KEMP_D$ on $D$ causes the equilibrium concentration of SVOC in the particle phase to be very low for small particles, which decreases the reaction rate (Eq. 3). As a result, particle phase chemistry has little impact on the diameter growth rate or molecular composition below about 10 nm. Above 10 nm, DIMER mass starts to accumulate in the particle, causing the mass fractions of NVOC, sulfuric acid, and ammonia decrease.

   Figure 4a shows the mass fraction ratio of SVOC to NVOC as a function of particle diameter. Below 10

250 nm, where particle phase chemistry has minimal impact, the ratio in Figure 4a for particle phase reaction is equivalent to that in Figure 2 for partitioning alone. The plots in Figures 2 and 4a diverge above about 10 nm because the DIMER mass fraction increases, causing the NVOC mass fraction to decrease.

   Figure 4b shows the mass fraction ratio of DIMER to NVOC. Below about 10 nm, hardly any DIMER is produced. Above 10 nm, the DIMER to NVOC ratio increases approximately linearly with increasing $D$ as expected

255 for a volume-limited process relative to a surface-limited process. Figures 3 and 4 show, for the conditions studied, that particle size dependencies of the growth rate and composition above 10 nm are driven mainly by particle phase chemistry rather than partitioning alone. Figure 4b illustrates how the impact of an accretion reaction might be observed experimentally through organic molecular composition measurements. If a systematic increase in the concentration of non-volatile dimers and higher order oligomers is observed with increasing particle diameter, then

260 particle phase chemistry is likely to have contributed to the formation of these molecules.

### 3.3 Factors that Influence Growth by Particle Phase Chemistry

   Additional calculations were performed to investigate the roles of SVOC gas phase mixing ratio and particle phase reaction rate constant on growth rate and composition. In Figure S2, diameter growth rates are shown for SVOC gas phase mixing ratios between 0.35 to 1.4 pptv and compared to the partitioning calculation in Figure 1a (the "1.4 - NR" plot for no reaction in Figure S2). Above about 10 nm, the growth rates increase linearly with

265 increasing $D$, and the slope of the increase scales by approximately $[SVOC]^2$ as expected by Eq. 3. The actual dependence is slightly less than $[SVOC]^2$ owing to the change in particle density that is associated with the changing DIMER mass fraction.

   Figure 5a shows the effect of reaction rate constant on diameter growth rate. The rate constants investigated

270 range over two orders of magnitude from $10^{-3}$ to $10^{-1}$ $M^{-1}s^{-1}$ while keeping the SVOC mixing ratio constant at 1.4 pptv. "NR" in Figure 5a represents no reaction and is equivalent to the plot in Figure 1a for growth by partitioning alone. For a reaction rate constant of $10^{-3}$ $M^{-1}s^{-1}$, particle growth rate is only slightly larger than growth by

partitioning only. In this case, DIMER production is slow, so there is only minor enhancement of the overall growth rate. Increasing the rate constant one order of magnitude to $10^{-2}$ $M^{-1}s^{-1}$, causes a substantial enhancement of the

275 growth rate above 10 nm. For a rate constant of $10^{-1}$ $M^{-1}s^{-1}$, a complex plot is observed. Volume-limited particle growth occurs up to ~40 nm where the growth rate increases linearly with *D*. Above ~40 nm, the reaction rate becomes so fast that particle growth is limited by the condensation rate of SVOC, a surface-limited process, and the diameter growth rate and molecular composition become independent of *D*. Since the calculation methodology used in our study did not include an evaporation rate for SVOC, the transition from volume-limited to surface-limited

kinetics is abrupt in Figure 5a. In practice, one would expect a more gradual transition from one to the other. The lack of a particle size dependence on SOA growth in the limit of a fast reaction rate has also been suggested in a modeling study of SOA produced by α-pinene ozonolysis (Gatzsche et al., 2017). It is also consistent with the work of Vesterinen et al. (Vesterinen et al., 2007) who showed a linear increase of particle diameter with time when the reaction rate constant was sufficiently large.

Accretion chemistry is not necessarily restricted to SVOC molecules. If two NVOC molecules react in the particle phase to form DIMER, then the diameter growth rate *will not* change with increasing particle diameter since NVOC uptake remains unaffected, though the molecular composition *will* change with increasing diameter. A calculation demonstrating this principle is shown in Figure S3 (growth rate vs. particle diameter) and Figure S4 (DIMER/NVOC mass ratio vs. particle diameter). If one SVOC molecule reacts with one NVOC molecule to form

DIMER, both the growth rate and molecular composition change with increasing particle diameter. A calculation demonstrating this principle is shown in Figure 5b, where growth rate vs. particle diameter is compared for SVOC-SVOC, NVOC-NVOC and SVOC-NVOC dimer formation reactions. The SVOC-NVOC reaction enhances the particle growth rate at a much smaller particle diameter than the SVOC-SVOC reaction. This enhancement arises from $KEMP_D$ of SVOC, which favors formation of the SVOC-NVOC dimer over the SVOC-SVOC dimer at small

particle sizes. Because of the greater potential of the SVOC-SVOC reaction to transform semi-volatile matter into non-volatile matter, its contribution to growth rate eventually overcomes that of the SVOC-NVOC reaction as the particle size increases.

Taken together, Figures S2-S4 show that the observation of a change in DIMER to monomer ratio with increasing particle diameter does not necessarily indicate an enhancement of the particle growth rate by accretion

chemistry. On the other hand, Figure 5b shows that the lack of a particle size-dependent change in the growth rate does not necessarily mean that particle growth rate is unaffected by accretion chemistry. Nonetheless, these results illustrate how experimental measurements of particle size-dependent changes in molecular composition and growth rate can supplement traditional measures, such as aerosol yield and perturbations due to e.g. isothermal dilution, to better constrain SOA formation models with regard to particle phase chemistry.

**3.4 Comparison to Recent Experimental Measurements and Atmospheric Implications**

For the modeling conditions studied here, particle phase chemistry influences both the molecular composition and growth rate of sub-100 nm diameter particles under atmospherically relevant conditions, provided

that the rate constant is above about $10^{-3}$ $M^{-1}s^{-1}$ for semi-volatile reactants having a saturation concentration on the order of 1 μg/m$^3$ and gas phase mixing ratios on the low pptv level. Based on the condensed phase concentration dependence of reactants on the reaction rate, a rate constant on the order of $10^{-1}$ $M^{-1}s^{-1}$ would be required for semi-volatile reactants having saturation concentrations on the order of 10 μg/m$^3$. These rate constants (Ziemann and Atkinson, 2012) and saturation concentrations (Trump and Donahue, 2014) are in the range of those expected for e.g. monoterpene accretion chemistry. The overall conclusion that accretion chemistry may contribute to nanoparticle growth and/or composition under atmospherically relevant conditions is similar to that reached by (Vesterinen et al., 2007). The current work provides an additional, fundamental basis for interpreting recent experimental measurements of molecular composition as a function of particle size in the case of monodisperse aerosols (Tu and Johnston, 2017), or more generally, volume-to-surface area ratio for polydisperse aerosols (Wu and Johnston, 2017). In each study, the relative concentration of accretion products increased approximately linearly with particle size (or volume-to-surface area ratio) as predicted by Figure 4b. Below, we estimate the magnitudes of the reaction rate constants needed to reproduce the experimental results.

In the Wu study (Wu and Johnston, 2017), decamethylcyclopentasiloxane (D$_5$) reacted with OH in the gas phase to produce secondary aerosol having three types of products: ring-opened species, oxidized monomers containing OH and/or CH$_2$OH functionalities in place of CH$_3$, and dimers of D$_5$ and/or its monomer oxidation products. The dimer concentration in the particle phase (as indicated by the signal intensities of corresponding ions produced by electrospray ionization) increased linearly with increasing volume-to-surface area ratio of the aerosol relative to the (non-volatile) ring-opened products. This dependence suggested that ring-opened products were formed in the gas phase and subsequently condensed onto the particles, while dimers were produced directly in the particle phase from monomers. Monomer gas phase mixing ratios in these experiments were on the order of a few ppbv, which is similar in magnitude to the levels measured in a university lecture hall (Tang et al., 2015). Using these values in conjunction with the basic model described above, we estimate the dimerization rate constant had to be in the $10^{-3}$ to $10^{-1}$ $M^{-1}s^{-1}$ range to achieve the level of dimerization observed, given that most of the monomers expected to participate in the reaction had estimated saturation concentrations in the $10^2$ to $10^3$ μg/m$^3$ range. Smaller rate constants would have been unable to produce a sufficient amount of dimers, while higher rate constants would not have shown a volume-to-surface area ratio dependence of the dimer concentration since the reaction rate would have been limited by transport of reacting monomer to the particle surface.

In the Tu study (Tu and Johnston, 2017), β-pinene ozonolysis produced SOA having systematic changes in molecular composition as a function of particle size. The relative concentrations (as indicated by the signal intensities of ions produced by electrospray ionization) of higher order oligomers, i.e. trimers and tetramers, increased linearly with increasing particle size, similar to Figure 4b. The total oligomer signal intensity was comparable to our previous study of α-pinene SOA formed under similar reaction conditions, where the oligomer content was experimentally determined to be about 50% of the total SOA mass (Hall IV and Johnston, 2011). For the Tu study, we estimate the NVOC mixing ratio to be approximately 2 ppbv based on the yield estimate of Ehn et al. (Ehn et al., 2014) for β-pinene ozonolysis. We estimate the mixing ratios for the 10 μg/m$^3$ and 100 μg/m$^3$

saturation concentration bins to be approximately 30 ppbv and 40 ppbv, respectively, based on the volatility basis set (VBS) parameterization of Donohue and coworkers for α-pinene ozonolysis (Donahue et al., 2012; Trump and Donahue, 2014). For these conditions, condensational growth and/or partitioning of molecular species alone are able to explain the overall growth of particles to about 80-100 nm in diameter during the ~20 s residence time of the flow reactor, though the modeling results are extremely sensitive to the mixing ratios and VBS parameterization used. However, condensational growth and partitioning cannot explain the high abundance of accretion reaction products that were measured. We estimate that the reaction rate constants needed to form dimers and higher order oligomers in sufficient amounts had to be at least on the order of $10^{-3}$ M$^{-1}$s$^{-1}$. (The required rate constant is dependent on SVOC volatility, with a larger rate constant needed for saturation concentrations above 10 μg/m$^3$.) Together, the modeling work performed here and the experimental measurements in the Wu and Tu studies all suggest that reaction rate constants on the order of $10^{-3}$ to $10^{-1}$ M$^{-1}$s$^{-1}$ are needed for accretion chemistry to be relevant to SOA molecular composition and growth.

While systematic changes in molecular composition with increasing particle size and/or aerosol volume-to-surface area ratio have been established experimentally, an enhancement of the particle growth rate in the sub-100 nm diameter range is less clear. Recent measurements by Kourtchev et al. (Kourtchev et al., 2016) of accretion oligomers in ambient aerosol from a boreal forest suggest that oligomer formation increases with increasing SOA mass concentration, an observation that is consistent with the molecular composition results reported here. More importantly, the authors noted that aerosols enriched with oligomers were strongly correlated with higher CCN activity, and they suggested that this correlation could indicate that oligomers may speed up particle growth. The modeling results presented here suggest that particle phase reactions are indeed capable of enhancing growth rates in a size range relevant to CCN activity.

The work presented here suggests that experimental measurements of molecular composition as a function of particle size and/or aerosol volume-to-surface area ratio can supplement traditional measurements (aerosol mass yields, perturbations caused by isothermal dilution, etc.) for constraining aerosol formation models. Future modeling of size-dependent molecular composition should include physico-chemical parameters such as diffusion, phase separation, reaction reversibility, etc. (Liu et al., 2014; Mai et al., 2015; Riipinen et al., 2012; Shiraiwa et al., 2012; Song et al., 2015; Trump and Donahue, 2014; Zaveri et al., 2014) that may influence the contribution of particle phase chemistry to nanoparticle growth.

**Supporting Information**

Four figures (S1-S4) plus additional description of the particle growth model.

**Acknowledgements**

This research was supported by the National Science Foundation under grant numbers CHE-1408455 and AGS-1649719.

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

**Tables and Figures**

**Table 1:** Mixing ratios and relevant physical properties of chemical species included in particle growth calculations. (T = 282 $^{o}$C, RH = 50%)

| | Mixing Ratio, $C_{i,g}$ | | Saturation Concentration | Molar Mass | Density |
|---|---|---|---|---|---|
| | ng m$^{-3}$ | pptv | log C*, (µg m$^{-3}$) | g mol$^{-1}$ | g cm$^{-3}$ |
| **Sulfuric Acid** | 0.5 | 0.12 | - | 98 | 1.8 |
| **Ammonia** | 1 | 1.25 | - | 17 | 0.7 |
| **SVOC** | 3-11 | 0.4-1.4 | $10^{0}$ | 200 | 1.2 |
| **NVOC** | 3 | 0.4 | $10^{-4}$ | 200 | 1.2 |
| **DIMER** | - | - | - | 400 | 1.2 |

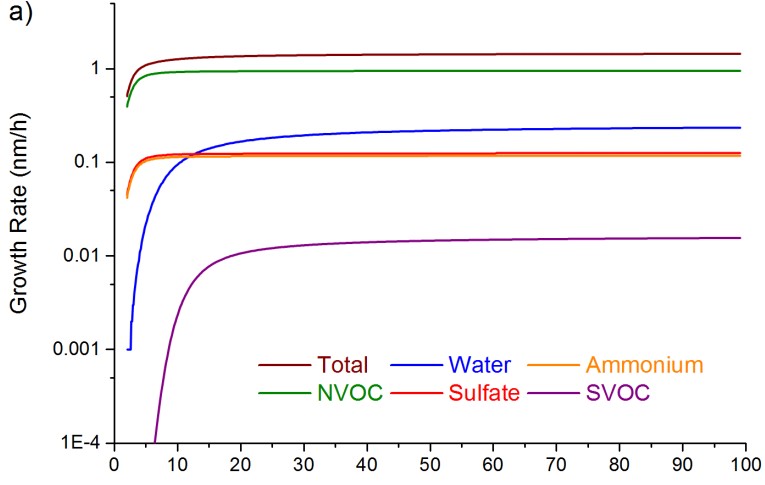

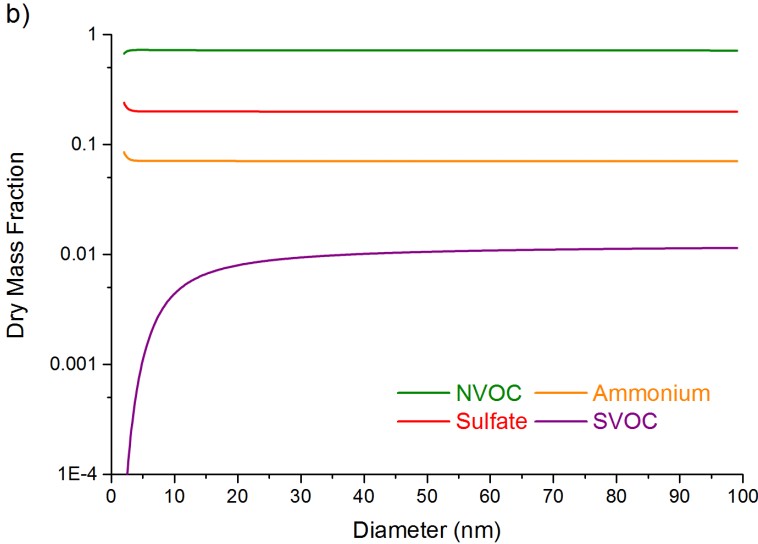

**Figure 1:** Particle diameter dependence of a) growth rates and b) dry mass fractions of chemical species under the
conditions where the gas phase mixing ratios are constant and growth occurs by partitioning alone.

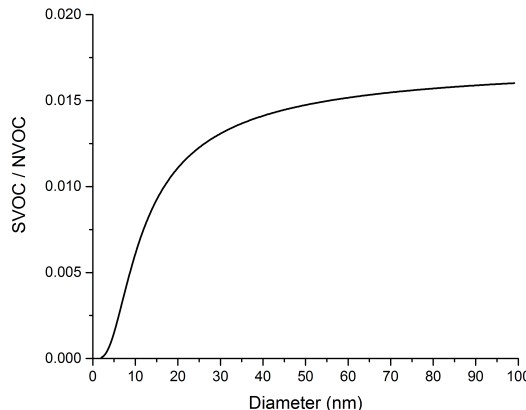

**Figure 2:** Mass fraction ratio of SVOC / NVOC vs. particle diameter under conditions where growth occurs by partitioning alone.

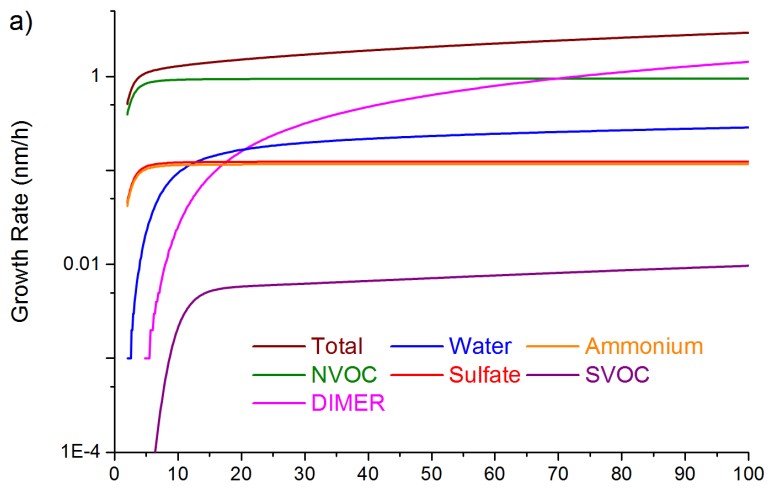

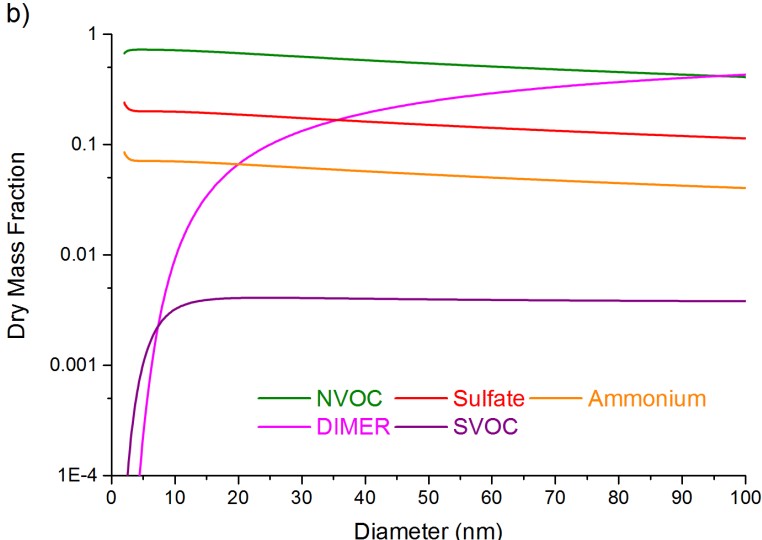

**Figure 3:** Particle diameter dependence of a) growth rates and b) dry mass fractions of chemical species under the conditions where both partitioning and particle phase chemistry (SVOC-SVOC dimer formation) are included.

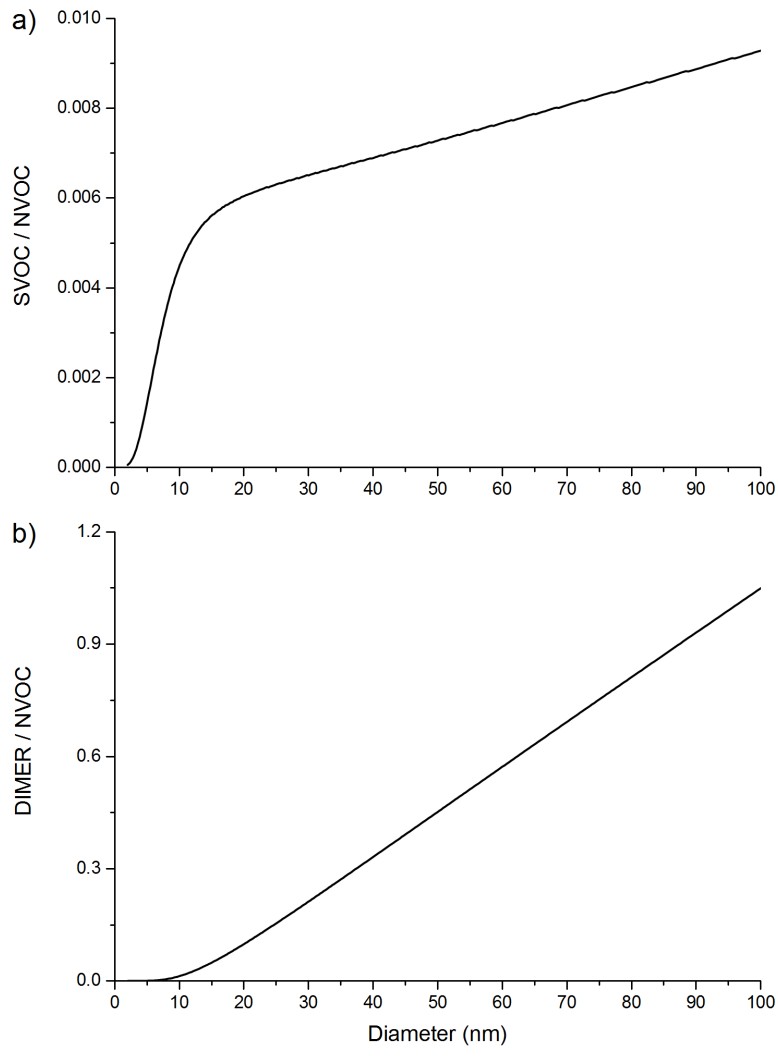

**Figure 4:** Mass ratio vs. particle diameter for a) SVOC to NVOC and b) DIMER to NVOC, under conditions where

both partitioning and particle phase chemistry (SVOC-SVOC dimer formation) are included.

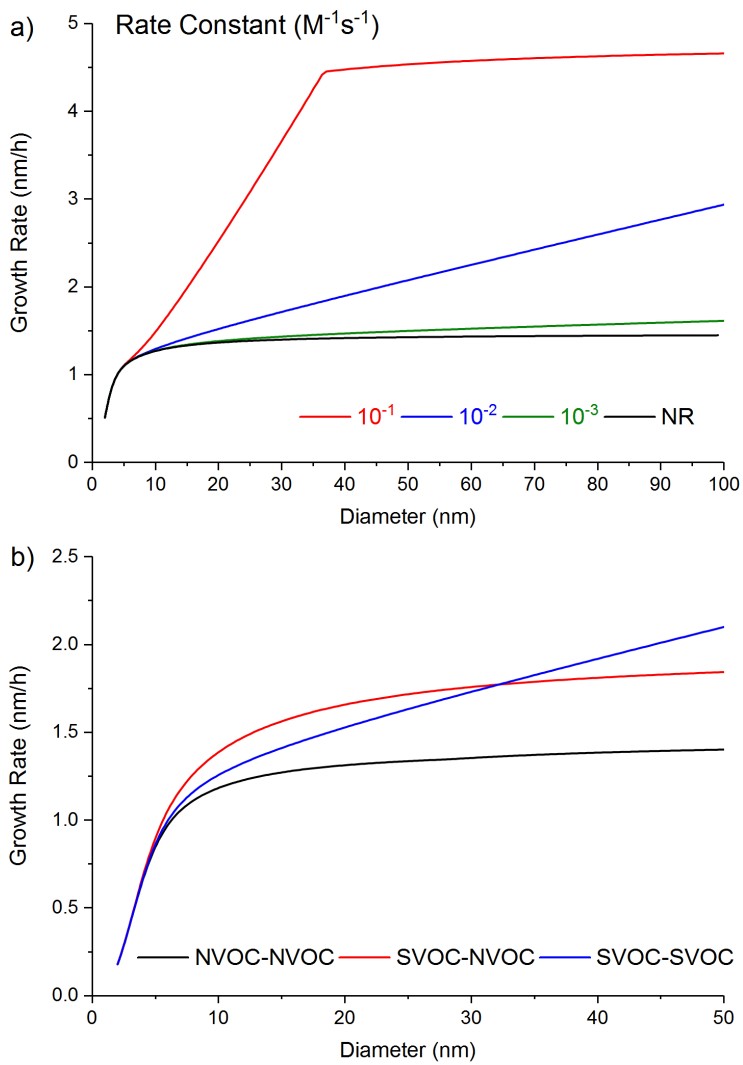

**Figure 5:** Growth rate vs. particle diameter:  a) Comparison of growth rates for different DIMER formation rate

constants in the SVOC-SVOC reaction.  b) Comparison of growth rates for SVOC-SVOC, NVOC-NVOC and

SVOC-NVOC reactions having a DIMER formation rate constant of $10^{-2}$ $M^{-1}s^{-1}$.  In these plots, the black line is

equivalent to the particle diameter dependence in Figure 1a.