# Peer review of "Nanoparticle Growth by Particle Phase Chemistry"

_Atmospheric Chemistry and Physics, 2017_

## Referee Comment (RC1) · Anonymous Referee #2 · 1 Sep 2017

The Apsokardu and Johnston manuscript presents a modeling analysis of nanoparticle growth by two pathways involving organic compounds. One pathway is the irreversible condensation of low-volatility organic compounds (NVOCs) and the other is the reactive uptake of semi-volatile organic compounds (SVOCs). In this study, reactive uptake is driven by the particle-phase formation of a non-volatile dimer from a model SVOC. While not particularly complex or novel, the study nicely illustrates the differences in modeled particle-phase composition and growth rates (including the dependencies on particle diameter and volume) when an accretion reaction is considered. It should be a useful study for interpretation of past and future field measurements of growth rates of 10-100 nm particles, particularly when coupled with composition data. The paper is logically written and includes appropriate figures and references. It is suitable for

publication in ACP. Minor comments are provided below.

1. What was the motivation for choosing the reaction rate constant? Dimerization of glyoxal into a bulk aqueous solution does not seem to be a particularly relevant reaction for considering accretion reactions in small particles. Measured and calculated rate constants for various accretion reactions, likely more relevant, have been reported by the Johnston group (De Palma et al.) and Ziemann and Atkinson (Chem. Soc. Rev. 2012). At a minimum, it is suggested that this value be placed in some context and a reference be made to the latter sensitivity studies (section 3.3). 2. The authors may wish to review a recent publication by Mohr et al. (GRL 2017) that models the contribution of dimers to new particle formation. The dimers are formed in the gas-phase, but the paper may nonetheless have some relevance for this work.

---

## Author Comment (AC1) · 14 Sep 2017

The authors thank the referee for helpful comments to strengthen and clarify the manuscript. Our responses are provided below.

1. The referee would like clarification on our motivation for the selection of dimerization rate constants used in our calculations.

Author response: Our calculations used dimerization rate constants in the range 10-3 to 10-1 M-1s-1 with most calculations at 10-2 M-1s-1. As stated in the original manuscript, 10-2 M-1s-1 happens to be the reported value for dimerization of glyoxal. The reference cited by the referee (Ziemann, P. J. and Atkinson, R., Chem. Soc. Rev., 41, 6582–6605, 2012) reviews kinetic and thermodynamic data for several types

of accretion reactions. Reactions relevant to growth of biogenic SOA are the reaction of a hydroperoxide with a carbonyl to give a peroxyhemiacetal, and the reaction of a peroxyacid with a carbonyl to form an acyl peroxyhemiacetal. In both cases, rate constants are reported to be in the $10^{-4}$ to $10^{-2}$ $M^{-1}s^{-1}$ range. Reactions such as aldol condensation of carbonyls and ester formation from an acid and alcohol are unlikely to be atmospherically relevant based on both kinetics (Casale, M.T., et al., Atmos. Environ., 41, 6212–6224, 2007) and thermodynamics (DePalma, J.W. et al., Phys. Chem. Chem. Phys., 15, 6935-6944, 2013). In the revised manuscript, we will add the Ziemann and Atkinson reference and expand our discussion of the relevance of the rate constants we used for these calculations.

2. The referee suggests that the authors review a recent publication that focuses on gas-phase dimer formation (Mohr, C. et al. Geophys. Res. Lett., 44, 2958-2966, 2017) for possible relevance to the manuscript.

Author response: The authors acknowledge the importance of Mohr et. al., 2017 for understanding biogenic SOA formation and growth. Dimer formation in the gas phase is relevant to our work since it provides a mechanism for producing nonvolatile organic compounds (NVOC) that can subsequently condense onto particles. Because we did not model NVOC formation in our work, but simply used a relevant gas phase mixing ratio (0.4 pptv), we have not referenced the Mohr paper. However, we expect this paper will be widely read and cited by the scientific community.

---

## Referee Comment (RC2) · Anonymous Referee #1 · 22 Sep 2017

"Nanoparticle growth by particle phase chemistry," by Apsokardu and Johnston, describes a model in which ultrafine aerosol particle growth is represented by a combination of condensation and particle-phase reactions (here modeled as dimer formation). The ability to model the growth of nucleated particles to sizes that allow them to become cloud condensation nuclei is necessary in order to assess the potential impacts of new particle formation on cloud properties, and thus climate. Because of this, the subject of this manuscript is of value to the atmospheric chemistry community and appropriate to publish in ACP.

I guess my first reaction upon reading this paper is one of déjà vu. The authors devote no text to describing the rich history of modeling the growth of particles from a few nanometers to CCN-relevant sizes, but in fact there have been several papers devoted

to this. The one that is most similar to the current manuscript is a paper by Vesterinen et al., entitled "Effect of particle phase oligomer formation on aerosol growth" (https://doi.org/10.1016/j.atmosenv.2006.10.024). From what I can tell – and admittedly I didn't spend nearly as much time reading this paper as the authors should – the model described in that 2007 paper does an excellent job with the organics and particle-phase reactions, with the added benefits that the study by Vesterinen et al. (a) provides a sensitivity study of the effect of the dimer formation rate constant; (b) compares modeling results to chamber studies; (c) includes the rate of monomer formation and, I think, includes the possibility that SVOC may evaporate; and (d) provides a hypothesis for uptake that involves the formation of an organic liquid layer onto an ammonium sulfate seed. The main difference between the two studies is that the current one includes the uptake of ammonia and sulfuric acid, whereas the earlier one assumes an ammonium sulfate seed particle. Here is an opportunity for the authors to distinguish themselves from this earlier work, but in doing so they need to acknowledge the prior work and provide an explanation for why this builds upon existing models and how their results compare. There are likely other attempts at modeling growth to CCN size, but the reader would not get a sense that anything has been done in this area considering the scarcity of discussion of prior work by the authors.

In addition, the authors provide a fairly terse interpretation of the modeling results and no quantitative comparisons of their results to observations. In my view this is a major weakness of the manuscript. The system under study – ultrafine aerosol particle growth from terpene oxidation and on ammonium sulfate seed particles, is one of the most widely studied chemical systems, both in lab and in the field. The fact that no data are directly compared to modeling results is, in my view, a missed opportunity that does nothing to validate the assumptions that went into the development of the model.

In summary, this manuscript should acknowledge prior studies and discuss how their results both compare to and improve upon understanding of those earlier models. In

addition, some comparison of lab or field data seems like a reasonable thing to do for any new model presented to the community. At the very least, the authors could perform a calculation similar to that of Vesterinen et al. and turn off ammonia and sulfuric acid and study organic uptake and particle-phase reaction chemistry onto ammonium sulfate seed particles.

Minor points:

Title: Since the size range covered here is sub-100 nm diameters, then it would seem natural to use the accepted term "ultrafine aerosol particle" rather than "nanoparticle."

The authors cite their own studies often exclusively when several other seminal studies have contributed significantly to developing current understanding. Examples include "Nanoparticle composition and growth rate are dominated by organic matter," and the "growth rate of nanoparticles by sulfuric acid and base is predicted by condensational growth model," both of which had a history of important breakthroughs prior to the work of Bzdek and Pennington (here are just a few):

Nanoparticle growth from sulfuric acid and ammonia: http://onlinelibrary.wiley.com/doi/10.1029/2005JD005935/abstract

Nanoparticle growth from sulfuric acid and organics: http://onlinelibrary.wiley.com/doi/10.1029/2005GL023827/pdf

Nanoparticle growth from organics: http://onlinelibrary.wiley.com/doi/10.1029/2007GL032523/full

---

## Author Response (AR1)

**Nanoparticle Growth by Particle Phase Chemistry**

**Michael J. Apsokardu and Murray V. Johnston**

mvj@udel.edu

We thank the referees for their time and effort in this review, and especially for their comments to help clarify, expand, and strengthen the manuscript. Referee comments are given in **bold print**. Author responses are in normal font, and the section/paragraph/line numbers refer to the revised manuscript. A red line version of the manuscript showing changes from the original manuscript is appended to the end of this file. We note that the red line version (produced in MS Word) does not indicate the literature citations that have been added. These citations are included in the comments below.

**Response to the comments of Referee #1**

The authors thank the referee for significant and quite helpful comments. We have made several changes to the manuscript to address this referee's concerns.

**Referee comment #1:**

**I guess my first reaction upon reading this paper is one of déjà vu. The authors devote no text to describing the rich history of modeling the growth of particles from a few nanometers to CCN-relevant sizes, but in fact there have been several papers devoted to this. The one that is most similar to the current manuscript is a paper by Vesterinen et al., entitled "Effect of particle phase oligomer formation on aerosol growth" (https://doi.org/10.1016/j.atmosenv.2006.10.024). From what I can tell – and admittedly I didn't spend nearly as much time reading this paper as the authors should – the model described in that 2007 paper does an excellent job with the organics and particle-phase reactions, with the added benefits that the study by Vesterinen et al. (a) provides a sensitivity study of the effect of the dimer formation rate constant; (b) compares modeling results to chamber studies; (c) includes the rate of monomer formation and, I think, includes the possibility that SVOC may evaporate; and (d) provides a hypothesis for uptake that involves the formation of an organic liquid layer onto an ammonium sulfate seed. The main difference between the two studies is that the current one includes the uptake of ammonia and sulfuric acid, whereas the earlier one assumes an ammonium sulfate seed particle. Here is an opportunity for the authors to distinguish themselves from this earlier work, but in doing so they need to acknowledge the prior work and provide an explanation for why this builds upon existing models and how their results compare. There are likely other attempts at modeling growth to CCN size, but the reader would not get a sense that anything has been done in this area considering the scarcity of discussion of prior work by the authors.**

Author response #1:

Thank you very much for this comment. We are aware of the Vesterinen paper and are cited in some of our previous papers as well as other modeling work related to oligomers. We have addressed this comment in several ways.

First, we have added the following paragraph to section 1 lines 70-79 to briefly review modeling of oligomers in secondary aerosol formation, citing a total of 6 new publications and a seventh one that was already cited elsewhere in the manuscript.

"The role of particle phase oligomerization in SOA formation has been the focus of several modeling studies. Owing to the high molecular weight and corresponding low volatility of oligomer products (Shiraiwa et al., 2014), early work assumed an irreversible process (Vesterinen et al., 2007), which proved effective for predicting the yields of freshly formed aerosol in chamber experiments and estimating the magnitude of the oligomerization rate constant needed for the process to impact yields. More recent models have included reversibility (Roldin et al., 2014; Trump and Donahue, 2014), which is needed to reproduce perturbations of freshly formed SOA such as changes induced by isothermal dilution, thermal degradation and/or aging. Regional air quality models show that oligomerization has the potential to significantly increase the SOA mass concentration (Aksoyoglu et al., 2011; Lemaire et al., 2016), and accurately representing this chemistry in these models is perhaps the greatest uncertainty for predicting SOA formation (Shrivastava et al., 2016)."

Second, we made additions to the manuscript in several places to indicate the "value added" aspect of our study relative to the body of work summarized in the new paragraph above. These include:

(1) Our manuscript explicitly discusses particle size effects in the sub-100 nm diameter regime. To address this point, we made the following addition to section 1 lines 80-85:

"The influence of particle size, or more precisely the relative roles of particle volume and surface area, on oligomer formation in SOA has received relatively little attention though these effects are implicit in all of the models. Particle size has been discussed primarily in the context of accumulation mode particles greater than 100 nm in diameter (Roldin et al., 2014; Shiraiwa et al., 2013; Vesterinen et al., 2007). The Vesterinen study did report aerosol yield functions from simulations starting with 20 nm diameter seed particles, and the results suggested that particle phase chemistry could enhance growth rates under atmospherically relevant conditions."

(2) Our modeling parameters more closely resemble those in a flow reactor than a batch reactor, which was the main focus of other modeling studies of laboratory data. This becomes important when we discuss modeling of recent experiments performed in a flow tube (see response to comment 2 of referee #1).

We made the following addition to final paragraph of section 1 lines 95-99: "The modeling approach is similar to that used in previous studies, though the reaction conditions studied here more closely resemble those of an "open" laboratory reactor where aerosol flows into and out of the reactor (i.e. flow reactor), as opposed to a "closed" batch reactor which was the main focus of previous modeling work. The results are discussed in the context of recent

size-resolved molecular composition measurements (performed with flow reactors) and the potential atmospheric impact of particle phase chemistry."

We also made the following addition to end of first paragraph of section 2 lines 110-112: "The gas phase mixing ratios are assumed to be constant over time (steady state) as might be achieved in a flow reactor, and the values chosen are typical of what might be observed during NPF in a boreal forest (Vestenius et al., 2014). "

(3) Our manuscript presents a detailed discussion of particle size-dependent molecular composition, including a discussion of recent experimental results (see response to comment 2 of referee #1). While particle composition is an implicit aspect of previous studies, it has not been explicitly discussed in detail. For example, plots similar to Figures 3 and 4 simply do not exist in the literature nor have they been discussed in that manner. To address this topic, we revised the first paragraph of section 3.4 lines 306-320 as follows:

"For the modeling conditions studied here, particle phase chemistry influences both the molecular composition and growth rate of sub-100 nm diameter particles under atmospherically relevant conditions, provided that the rate constant is above about $10^{-3}$ $M^{-1}s^{-1}$ for semi-volatile reactants having a saturation concentration on the order of 1 $\mu g/m^3$ and gas phase mixing ratios on the low pptv level. Based on the condensed phase concentration dependence of reactants on the reaction rate, a rate constant on the order of $10^{-1}$ $M^{-1}s^{-1}$ would be required for semi-volatile reactants having saturation concentrations on the order of 10 $\mu g/m^3$. These rate constants (Ziemann and Atkinson, 2012) and saturation concentrations (Trump and Donahue, 2014) are in the range of those expected for e.g. monoterpene accretion chemistry. The overall conclusion that accretion chemistry may contribute to nanoparticle growth and/or composition under atmospherically relevant conditions is similar to that reached by (Vesterinen et al., 2007). The current work provides an additional, fundamental basis for interpreting recent experimental measurements of molecular composition as a function of particle size in the case of monodisperse aerosols (Tu and Johnston, 2017), or more generally, volume-to-surface area ratio for polydisperse aerosols (Wu and Johnston, 2017). In each study, the relative concentration of accretion products increased approximately linearly with particle size (or volume-to-surface area ratio) as predicted by Figure 4b. Below, we estimate the magnitudes of the reaction rate constants needed to reproduce the experimental results."

(4) To further drive home the emphasis on size-dependent particle composition, we have added a new figure and text showing how the identities of the monomer reactants affect particle composition and growth. To the authors' knowledge, there are no other reports in the literature modeling the relative roles of SVOC and NVOC in accretion chemistry; most simply discuss accretion chemistry in terms of SVOC alone. Accordingly we have revised the last two paragraphs in section 3.3 lines 285-304 and added a new Figure 5b:

"Accretion chemistry is not necessarily restricted to SVOC molecules. If two NVOC molecules react in the particle phase to form DIMER, then the diameter growth rate *will not* change with increasing particle diameter since NVOC uptake remains unaffected, though the molecular composition *will* change with increasing diameter. A calculation demonstrating this principle is shown in Figure S3 (growth rate vs. particle diameter) and Figure S4 (DIMER/NVOC mass ratio vs. particle diameter). If one SVOC molecule reacts with one NVOC molecule to form

DIMER, both the growth rate and molecular composition change with increasing particle diameter. A calculation demonstrating this principle is shown in Figure 5b, where growth rate vs. particle diameter is compared for SVOC-SVOC, NVOC-NVOC and SVOC-NVOC dimer formation reactions. The SVOC-NVOC reaction enhances the particle growth rate at a much smaller particle diameter than the SVOC-SVOC reaction. This enhancement arises from $KEMP_D$ of SVOC, which favors formation of the SVOC-NVOC dimer over the SVOC-SVOC dimer at small particle sizes. Because of the greater potential of the SVOC-SVOC reaction to transform semi-volatile matter into non-volatile matter, its contribution to growth rate eventually overcomes that of the SVOC-NVOC reaction as the particle size increases.

Taken together, Figures S2-S4 show that the observation of a change in DIMER to monomer ratio with increasing particle diameter does not necessarily indicate an enhancement of the particle growth rate by accretion chemistry. On the other hand, Figure 5b shows that the lack of a particle size-dependent change in the growth rate does not necessarily mean that particle growth rate is unaffected by accretion chemistry. Nonetheless, these results illustrate how experimental measurements of particle size-dependent changes in molecular composition and growth rate can supplement traditional measures, such as aerosol yield and perturbations due to e.g. isothermal dilution, to better constrain SOA formation models with regard to particle phase chemistry."

(5) Finally, we added the following text to the end of Section 3.3 lines 365-371 that succinctly summarizes the most important "value added" aspect of the work we report here: "The work presented here suggests that experimental measurements of molecular composition as a function of particle size and/or aerosol volume-to-surface area ratio can supplement traditional measurements (aerosol mass yields, perturbations caused by isothermal dilution, etc.) for constraining aerosol formation models. Future modeling of size-dependent molecular composition should include physico-chemical parameters such as diffusion, phase separation, reaction reversibility, etc. (Liu et al., 2014; Mai et al., 2015; Riipinen et al., 2012; Shiraiwa et al., 2012; Song et al., 2015; Trump and Donahue, 2014; Zaveri et al., 2014) that may influence the contribution of particle phase chemistry to nanoparticle growth."

**Referee comment #2.**

**In addition, the authors provide a fairly terse interpretation of the modeling results and no quantitative comparisons of their results to observations. In my view this is a major weakness of the manuscript. The system under study – ultrafine aerosol particle growth from terpene oxidation and on ammonium sulfate seed particles, is one of the most widely studied chemical systems, both in lab and in the field. The fact that no data are directly compared to modeling results is, in my view, a missed opportunity that does nothing to validate the assumptions that went into the development of the model.**

**In summary, this manuscript should acknowledge prior studies and discuss how their results both compare to and improve upon understanding of those earlier models. In addition, some comparison of lab or field data seems like a reasonable thing to do for any new model presented to the community. At the very least, the**

**authors could perform a calculation similar to that of Vesterinen et al. and turn off ammonia and sulfuric acid and study organic uptake and particle-phase reaction chemistry onto ammonium sulfate seed particles.**

Author response #2.

To address this comment, we renamed Section 3.4 "Comparison to Recent Experimental Measurements and Atmospheric Implications", and we added three new paragraphs in lines 306-355 to describe the use of the model to provide semi-quantitative insight into the experimental results of Wu and Johnston (2017) and Tu and Johnston (2017). Specifically, in the revised manuscript we estimate numerical values for dimerization rate constants that explain the experimental results along with their atmospheric relevance. Here are the paragraphs added:

[revised manuscript text omitted]

With regard to the reviewer suggestion to perform a calculation similar to Vesterinen (2007), we decided instead to add the following sentence to lines 280-283 of section 3.3, which directly compares an aspect of our results in Figure 5 (the $10^{-1}$ $M^{-1}s^{-1}$ calculation in this figure) to the Vesterinen work:  “The lack of a particle size dependence on SOA growth in the limit of a fast reaction rate has also been suggested in a modeling study of SOA produced by α-pinene ozonolysis (Gatzsche et al., 2017).  It is also consistent with the work of Vesterinen et al. (Vesterinen et al., 2007) who showed a linear increase of particle diameter with time when the reaction rate constant was sufficiently large.”

We also added the following to section 3.3 lines 313-315:  “The overall conclusion that accretion chemistry may contribute to nanoparticle growth and/or composition under atmospherically relevant conditions is similar to that reached by (Vesterinen et al., 2007).”

Additional Author response to comments 1 and 2 of referee #1:

The abstract has been revised (lines 6-20) to reflect changes to the manuscript discussed above. Here is the revision:

"The ability of particle phase chemistry to alter the molecular composition and enhance the growth rate of nanoparticles in the 2-100 nm diameter range is investigated through the use of a growth model. The molecular components included are sulfuric acid, ammonia, water, a non-volatile organic compound, and a semi-volatile organic compound. Molecular composition and growth rate are compared for particles that grow by partitioning alone vs. those that grow by a combination of partitioning and an accretion reaction in the particle phase between two organic molecules. Particle phase chemistry causes a change in molecular composition that is particle diameter dependent, and when the reaction involves semi-volatile molecules, the particles grow faster than by partitioning alone. These effects are most pronounced for particles larger than about 20 nm in diameter. The modeling results provide a fundamental basis for understanding recent experimental measurements of the molecular composition of secondary organic aerosol showing that accretion reaction product formation increases linearly with increasing aerosol volume-to-surface area. They also allow initial estimates of the reaction rate constants for these systems. For secondary aerosol produced by either OH oxidation of the cyclic dimethylsiloxane ($D_5$) or ozonolysis of $\beta$-pinene, oligomerization rate constants on the order of $10^{-3}$ to $10^{-1}$ $M^{-1}s^{-1}$ are needed to explain the experimental results. These values are consistent with previously measured rate constants for reactions of hydroperoxides and/or peroxyacids in the condensed phase."

**Referee comment #3: (minor points)**

**Title: Since the size range covered here is sub-100 nm diameters, then it would seem natural to use the accepted term "ultrafine aerosol particle" rather than "nanoparticle."**

**The authors cite their own studies often exclusively when several other seminal studies have contributed significantly to developing current understanding. Examples include "Nanoparticle composition and growth rate are dominated by organic matter," and the "growth rate of nanoparticles by sulfuric acid and base is predicted by condensational growth model," both of which had a history of important breakthroughs prior to the work of Bzdek and Pennington (here are just a few): Nanoparticle growth from sulfuric acid and ammonia: http://onlinelibrary.wiley.com/doi/10.1029/2005JD005935/abstract Nanoparticle growth from sulfuric acid and organics: http://onlinelibrary.wiley.com/doi/10.1029/2005GL023827/pdf Nanoparticle growth from organics: http://onlinelibrary.wiley.com/doi/10.1029/2007GL032523.**

Author response #3:

Title: We have also gone back and forth on the use of "ultrafine" vs. "nanoparticle" in the title. We have stuck with "nanoparticle" because we consider the full size range down to 2 nm, and in much of the scientific literature, the nomenclature is such that "nanoparticle" is used for the entire size range below 100 nm.

Citations: We have trimmed back the Bzdek/Pennington citations and added others. Specifically, paragraph 2 of section 1 lines 36-45 has been modified in the following way:

"The three main chemical species that contribute to ambient nanoparticle growth are sulfuric acid, a neutralizing base typically ammonia, and organic matter. The growth rate due to sulfuric acid along with neutralizing base is accurately predicted by experimental measurements of gas phase mixing ratio and particle phase composition using a condensational growth model (Bzdek et al., 2013; Pennington et al., 2013; Smith et al., 2008; Stolzenburg et al., 2005), though sulfuric acid represents only a minor fraction of the total growth rate of ambient particles (Kuang et al., 2010, 2012; Weber et al., 1996; Wehner et al., 2005). Nanoparticle composition and growth rate are dominated by organic matter (Bzdek et al., 2011, 2012, 2013, 2014a, 2014b; Pennington et al., 2013; Riipinen et al., 2012; Smith et al., 2008), and though significant molecular insight has been gained (Bianchi et al., 2016; Ehn et al., 2014; Kulmala et al., 2013; Riccobono et al., 2014), current growth models for organic matter appear to be incomplete (Hallquist et al., 2009; Tröstl et al., 2016)."

We have also modified references elsewhere in the manuscript (paragraph 3 of section 1; paragraph 1 of section 2) to emphasize more broadly the contributions of others.

**Response to the comments of Referee #2**

The authors thank the referee for helpful comments to strengthen and clarify the manuscript. Our responses and changes to the manuscript are provided below.

**Referee comment #1:**

**What was the motivation for choosing the reaction rate constant? Dimerization of glyoxal into a bulk aqueous solution does not seem to be a particularly relevant reactionfor considering accretion reactions in small particles. Measured and calculated rate constants for various accretion reactions, likely more relevant, have been reported by the Johnston group (De Palma et al.) and Ziemann and Atkinson (Chem. Soc. Rev. 2012). At a minimum, it is suggested that this value be placed in some context and a reference be made to the latter sensitivity studies (section 3.3).**

Author response #1:

Our calculations used dimerization rate constants in the range $10^{-3}$ to $10^{-1}$ $M^{-1}s^{-1}$ with most calculations at $10^{-2}$ $M^{-1}s^{-1}$. As stated in the original manuscript, $10^{-2}$ $M^{-1}s^{-1}$ happens to be the reported value for dimerization of glyoxal. The reference cited by the referee (Ziemann, P. J. and Atkinson, R., Chem. Soc. Rev., 41, 6582–6605, 2012) reviews kinetic and thermodynamic data for several types of accretion reactions. We have modified the final paragraph of section 2.2 lines 182-192 to address the question of choice of reaction rate constant:

"Dimerization rate constants in the range of $10^{-3}$ to $10^{-1}$ $M^{-1}$ $s^{-1}$ were studied, with most calculations at $10^{-2}$ $M^{-1}s^{-1}$, which is the rate constant reported for dimerization of glyoxal in a bulk aqueous solution (Ervens and Volkamer, 2010). Ziemann and Atkinson (Ziemann and Atkinson, 2012) have reviewed thermodynamic and kinetic data for several types of reactions relevant to biogenic SOA. The reaction of a hydroperoxide with a carbonyl to give a

peroxyhemiacetal, and the reaction of a peroxyacid with a carbonyl to form an acyl peroxyhemiacetal, both have reported rate constants in the $10^{-4}$ to $10^{-2}$ $M^{-1}s^{-1}$ range depending on reaction conditions, and are relevant to the modeling results presented here. Reactions such as aldol condensation of carbonyls and ester formation from an acid and alcohol are much slower and unlikely to be atmospherically relevant based on both kinetics (Casale et al., 2007) and thermodynamics (DePalma et al., 2013). The effects of dimer decomposition (reaction reversibility), particle phase diffusion coefficient, and phase separation are not considered in this work, though we note that all would have the effect of reducing the impact of particle phase chemistry on composition and growth rate."

**Referee comment #2:**

**The authors may wish to review a recent publication by Mohr et al. (GRL 2017) that models the contribution of dimers to new particle formation. The dimers are formed in the gas phase, but the paper may nonetheless have some relevance for this work.**

Author response #2:

The authors acknowledge the importance of Mohr et. al., 2017 for understanding biogenic SOA formation and growth. Dimer formation in the gas phase is relevant to our work since it provides a mechanism for producing nonvolatile organic compounds (NVOC) that can subsequently condense onto particles. Because we did not model NVOC formation in our work, but simply used a relevant gas phase mixing ratio (0.4 pptv), we have not referenced the Mohr paper. However, we expect this paper will be widely read and cited by the scientific community.

**Nanoparticle Growth by Particle Phase Chemistry**

Michael J. Apsokardu and Murray V. Johnston

[revised manuscript text omitted]

---

## Referee Report (RR1)

Final reviewer comments on "Nanoparticle Growth by Particle Phase Chemistry"

I feel the authors have fully addressed all of my initial concerns about this manuscript. Firstly, the authors have done a more thorough job reporting on the prior work in this research area as well as pointing out the unique contributions made by their work. In addition, the authors have compared model results with experimental studies from their own laboratory. This has resulted in further insights on model performance as well as additionally providing some constraints on dimerization rates in growing nanoparticles.

---

## Author Response (AR2)

Author Response:

We thank the referees and editor for their time and effort in reviewing this manuscript. We have made the changes as requested. In the time between submission of the revised manuscript and the current review, we found an error in the way we treated the particle size dependence of molecular transport to the particle surface (the beta term in Eq. 1). The impact of error is small and does not affect the conclusions of our work. It amounts to ~10% reduction of the growth rate for the largest particle diameter examined, and very little impact on the size-dependent molecular composition. Nonetheless, we are compelled to disclose the problem and rectify it in the manuscript. For this reason, we made additional changes to the text in the manuscript and supporting information, and we updated all figures even though the change was often imperceptible. As an aid to the editor/reviewers, we have appended redline documents to this file that highlight the changes that were made. Please contact us if you have questions or require additional information.

Murray Johnston
Michael Apsokardu

[revised manuscript text omitted]
$. Color shaded regions delineate particle growth by condensation (red), partitioning (green), and particle phase chemistry (yellow). Calculations begin with the Kelvin effect modified vapor pressure ($KEMP_d$):

$$(S1) \quad KEMP_d = P_0 e^{\left[(2\sigma V_{M,p})\,/\,\left(\frac{d}{2}RT\right)\right]}$$

where $P_0$ is the saturation vapor pressure over a flat surface, $\sigma$ is the surface tension, $V_{M,p}$ is the average molar volume of the particle, $d$ is the particle diameter, $R$ is the universal gas constant, and $T$ is the temperature. The subscript $d$, shown here with $KEMP_d$, and for other variables hereafter, represents the particle diameter and denotes that the variable is particle size dependent. The saturation ratio ($S_d$) is given by:

Murray Johnston 12/28/2017 1:10 PM

Murray Johnston 12/28/2017 1:10 PM

Murray Johnston 12/28/2017 1:10 PM

Murray Johnston 12/28/2017 1:10 PM

Murray Johnston 12/28/2017 1:10 PM

Murray Johnston 12/28/2017 1:10 PM

Murray Johnston 12/28/2017 1:10 PM

Murray Johnston 12/28/2017 1:10 PM

(S2)   $S_d = C_{I,g}/KEMP_{d_i}$

For compounds having $S_d < 1$, uptake occurs at a slower rate than the condensation rate, while for compounds having $S_d \gg 1$, uptake occurs at the condensation rate.

For the molecular species considered in this study, those growing the particle at the condensation rate (red shaded region of Figure S1) are sulfuric acid and non-volatile organic compound (NVOC). Equation 1 in the main text gives the uptake rate, which assumes that every collision results in uptake. The Loyalka mass transfer correction factor $\beta_d$ is given by:

(S3)   $\beta_d = \dfrac{\sqrt{\pi}\,Kn\,(1+1.333)}{1 + Kn\,(1.333 + 1.333\sqrt{\pi}\,Kn + \delta)}$

where $\alpha$ is the mass accommodation coefficient (assumed to be 1), $\delta$ is the mass transfer jump coefficient equal to

1.0161, and $Kn$ is the Knudsen number:

(S4)   $Kn = \dfrac{2\lambda}{d_p + d_i}$ , where $\lambda$ is the mean free path and $d$ is diameter. Subscripts $p$ and $i$ indicate particle and gas molecule respectively.

The mean free path is defined as:

(S5)   $\lambda = \dfrac{4\,(D_p + D_i)}{\sqrt{\pi}\,(c_p^2 + c_i^2)^{1/2}}$

where $D$ is the gas phase diffusion coefficient and $c$ is the mean thermal speed. The gas phase diffusion coefficient of a particle is defined as (S6)   $D_p = \dfrac{k_B\,T\,C_c}{3\,\pi\,\mu_{air}\,d_p}$

where Cc is the Cunningham slip correction factor and $\mu_{air}$ is the dynamic viscosity of air ($\mu_{air} = 1.76\times10^{-5}$ kg m$^{-1}$ s$^{-1}$

at 282 K). The gas phase diffusion coefficient of a vapor molecule is estimated from:

(S7)   $D_i = \dfrac{0.001\,T^{1.75}}{P\left(\Sigma_{air}^{1/3} + \Sigma_i^{1/3}\right)^2}\sqrt{\dfrac{1}{MW_{air}} + \dfrac{1}{MW_i}}$

where P is atmospheric pressure, $\Sigma$ is the atomic diffusion volume, and MW is the molecular weight. Eq. S7 is to calculate the gas phase diffusion coefficients for sulfuric acid and ammonia. For $\Sigma_{air}$ and $MW_{air}$, the values 19.7 and

28.97 g mol$^{-1}$ are used respectively. A list of atomic diffusion volumes can be found in Poling et al 2001. The diffusion coefficients for SVOC and NVOC vapor molecules were assumed to be 0.05 cm$^2$ s$^{-1}$. This value is consistent with Eq. S7 for typical molecular structures of monomer products of monoterpene ozonolysis.

Seinfeld and Pandis (Seinfeld, John H. and Pandis, Spyros N. "Atmospheric Chemistry and Physics: From

Air Pollution To Climate Change", John Wiley and Sons, New York, 2006, ISBN-13: 978-0471720188) give a detailed discussion of various calculation procedures for the mass transfer coefficient $\beta_d$ in Equation 1, including the

Murray Johnston 12/28/2017 1:10 PM

Murray Johnston 12/28/2017 1:10 PM

Murray Johnston 12/28/2017 1:10 PM

Murray Johnston 12/28/2017 1:10 PM

Murray Johnston 12/28/2017 1:10 PM

Murray Johnston 12/28/2017 1:10 PM

Murray Johnston 12/28/2017 1:10 PM

Murray Johnston 12/28/2017 1:10 PM

Murray Johnston 12/28/2017 1:10 PM

Murray Johnston 12/28/2017 1:10 PM

Murray Johnston 12/28/2017 1:10 PM

Murray Johnston 12/28/2017 1:10 PM

| Loyalka approach used in our work. As a check of our procedure, we also used the Fuchs-Sutugin approach to
| calculate $\beta_d$ and obtained equivalent results within the precision of the respective theories.

| Semi-volatile organic compounds (SVOC) cause particle growth at a rate that is slower than the
| condensation rate (green shaded region in Figure S1). Based on the gas phase mixing ratio and particle properties, a
| corresponding equilibrium particle phase concentration is calculated (Eq. 2 of the main text). The mass of such
| species added to the particle is based on $V_{p,n}$, so by $V_{p,n+1}$, the species is no longer in equilibrium and must be re-
| calculated. Partitioning of water is dependent on the mixing ratio and the $KEMP_d$ to determine the equilibrium mole
| fraction $x_i$. For simplicity, the activity coefficient ($\zeta$) is assumed to be 1.

| Particle phase chemistry occurs by an accretion reaction (yellow shaded region of Figure S1). Reactions are
| modeled by the second order decay of SVOC (or in the case of Figures S2 and S3, NVOC) to produce DIMER
| products. Depletion of SVOC is dependent on the concentration of SVOC existing in the particle at $V_{p,n}$. When
| equilibrium is re-calculated for SVOC at $V_{p,n+1}$, the mass added to the particle must account for both depletion by
| reaction and dilution due to particle growth. When the volume changes for all individual species have been
| calculated, they are summed to give the new particle volume, $V_{p,n+1}$. After volume $V_{p,n+1}$ is achieved, calculations are
| iteratively repeated.

| **Figure S2**

[Figure]

[Figure]

| Figure S2 shows growth rate vs particle diameter for the comparison of the growth rate due to particle
| phase reaction at different gas phase mixing ratios of SVOC. "1.4 - NR" indicates no particle phase reaction occurs
| and growth is only due to condensation and partitioning.

**Figure S3**

[Figure]

Figures S3a and b show the size dependent evolution of particle growth rate by a) partitioning alone and b)
with dimer formation from NVOC ($k_{II} = 10^{-3}$ M$^{-1}$s$^{-1}$).  Dimer formation from NVOC does not enhance the growth
rate (growth still proceeds at the NVOC collision rate), but it does change the composition.

**Figure S4**

[Figure]

Figure S4 shows the mass fraction ratio of DIMER to NVOC, which monotonically increases with
increasing particle diameter.

[Figure]

Murray Johnston 12/28/2017 1:10 PM

Murray Johnston 12/28/2017 1:10 PM

Murray Johnston 12/28/2017 1:10 PM